# Neuron Platonic Intrinsic Representation from Dynamics Using Contrastive Learning

**Wei Wu**
Peking University
`weiwu@stu.pku.edu.cn`

**Can Liao**
University of Georgia
`can.liao@uga.edu`

**Zizhen Deng**
Chinese Academy of Sciences
Institute of Automation
`18803835618@163.com`

**Zhengrui Guo**
The Hong Kong University of Science and Technology
Beijing Institute of Collaborative Innovation
`zguobc@connect.ust.hk`

**Jinzhuo Wang** ✉
Peking University
`wangjinzhuo@pku.edu.cn`

## Abstract

The Platonic Representation Hypothesis posits that behind different modalities of data (what we sense or detect), there exists a universal, modality-independent representation of reality. Inspired by this hypothesis, we treat each neuron as a system, where we can detect the neuron's multi-segment activity data under different peripheral conditions. We believe that, similar to the Platonic idea, there exists a time-invariant representation behind different segments of the same neuron, which reflects the intrinsic properties of the neuron's system. Intrinsic properties include the molecular profiles, location within brain regions and morphological structure, etc. The optimization objective for obtaining intrinsic neuronal representations should meet two criteria: (I) the representations of recording segments from the same neuron must exhibit higher similarity compared to those from different neurons; (II) the representations should generalize effectively to out-of-domain data. To this end, we propose the NeurPIR (Neuron Platonic Intrinsic Representation) framework, which leverages contrastive learning by treating segments from the same neuron as positive pairs and those from different neurons as negative pairs. In the implementation, we adopt VICReg, which only uses positive pairs while indirectly separating dissimilar samples through regularization terms. To validate the efficacy of our method, we first conducted tests on simulated neuronal population dynamics data generated by the Izhikevich model. The results confirmed that our approach accurately captured the neuron types as defined by the preset hyperparameters. Subsequently, we applied our method to two real - world neuron dynamics datasets, which included neuron type annotations derived from spatial transcriptomics and the location of each neuron within brain regions. The representations learned from our model not only accurately predicted neuron types and locations but also demonstrated robustness when tested on out-of-domain data (data from unseen animals). This finding underscores the potential of our approach in furthering the understanding of neuronal systems and offers valuable insights for future neuroscience research. Code is available at https://github.com/ww20hust/NeurPIR.

## 1 Introduction

Unraveling the intricacies of neuronal activity and the information encoded within neural dynamics stands as a monumental challenge in the field of neuroscience. Plato's cave allegory and Platonic Representation Hypothesis Huh et al. (2024) suggests the existence of a universal, modality-independent representation of world that transcends the modalities through which we perceive it. Drawing inspiration from this philosophical concept, we propose a novel perspective on neuronal

---

✉ : Corresponding Author

activity, treating each neuron as a distinct system capable of generating multi-segment activity data under various peripheral conditions, the multi-segment activity data of individual neuron as what we perceive.

Our approach, NeurPIR, is based on the premise that, similar to the Platonic idea, a time-invariant representation exists that underlies the diverse activity segments of the same neuron. This representation is hypothesized to encapsulate the intrinsic properties of the neuronal system, offering a stable and consistent framework for understanding neuronal function.

To extract intrinsic representations, we formulated an optimization objective based on two key criteria. First, activity segments from the same neuron should exhibit greater similarity than those from different neurons, ensuring that our method effectively captures the unique signatures of individual neurons. Second, the derived representations should demonstrate high generalizability, enabling their application to out-of-domain data, such as neuronal activity from different species or experimental conditions not included in the training set.

To achieve these objectives, we employed contrastive learning, a powerful machine learning technique that treats segments from the same neuron as positive pairs and those from different neurons as negative pairs. This approach leverages the contrast between similar and dissimilar samples to learn effective representations. For our implementation, we utilized VICReg, a variant of contrastive learning that focuses exclusively on positive pairs while incorporating regularization terms to indirectly separate dissimilar samples. This method ensures that the learned representations are both discriminative and robust to variations in the data. However, achieving the above requires the following two innovative designs for neuron data: Firstly, we use CEBRA Schneider et al. (2023b) to integrate the single neuronal peripheral information (such as activity of neighboring neuronal populations and behavioral data for each segment of a single neuron). This process encodes the peripheral information associated with each segment of an individual neuron. While CEBRA has been widely recognized for its ability to produce high-performance learnable latent embeddings for jointly representing surrounding neuronal data and external information, it was previously applied to groups of neurons rather than focusing on individual neurons. Secondly, the approach aims to produce a consistent representation of a neuron's identity across varying experimental conditions, using these conditions as positive instances in a contrastive learning paradigm. While multi-session data naturally provides such instances, the reality is that many neurons are recorded in only a single session. To address this limitation, we designed a data augmentation method tailored to neuronal data. This method involves extracting segments of varying lengths from the same session to serve as positive sample pairs. In this way, the learned representation can capture intrinsic properties across different time scales.

To rigorously assess the effectiveness of the representations learned by NeurPIR, we model neuron population dynamic data using the Izhikevich model Izhikevich (2003), where different neurons are assigned distinct hyperparameters representing different firing modes as intrinsic properties. These neurons are randomly connected to form a network, and after stimulating the network, we obtain neuron population data. The representations learned by through self-supervised learning on this neuron population data have been confirmed to contain hyperparameter information. In recognition of the noise and complexity in real-world neuronal datasets, we turn to a publicly available dataset of mouse brain neuron populations. These neurons, after recording dynamic activities, are separated and labeled with cell type annotations using transcriptomics. The representations learned by on this dataset can be efficiently utilized for downstream cell type prediction tasks. In addition, using another real public dataset containing ten mice and 39 datasets, we demonstrate that the intrinsic information contained in the representation obtained by is consistent across animals, and that this representation has strong cross-domain capability when performing downstream tasks.

## 2  RELATED WORK

**Single Neuron Models:** Single neuron models are essential in understanding the fundamental properties of neuronal dynamics and behavior. The Leaky Integrate-and-Fire model Liu & Wang (2001), developed in the 1950s, is a minimalistic model that simulates the integration of synaptic inputs and the leakage of membrane potential over time. The Hodgkin-Huxley model Nelson & Rinzel (1995), introduced in 1952, provides a detailed description of action potential generation using complex equations to represent ionic currents across the neuronal membrane, offering deep insights into neu-

ronal excitability. The FitzHugh-Nagumo model Izhikevich & FitzHugh (2006), proposed in 1961, simplifies the Hodgkin-Huxley model to focus on excitability and action potential dynamics while reducing computational complexity. Finally, the Izhikevich model, developed in 2003, combines simplicity with versatility, effectively capturing a wide range of neuronal firing patterns and balancing computational efficiency with biological realism. The hyperparameters set for these models can be viewed as prior assumptions about the inherent properties of the neurons. In the subsequent sections, we select the Izhikevich model to generate a synthetic dataset for testing , using the set hyperparameters as the true labels.

**Neural Latent Representation Learning:** Neural Latent Representation Learning has been pivotal in transforming high-dimensional neuronal data into lower-dimensional embeddings that encapsulate instantaneous information Bengio et al. (2013). A remarkable impact has been made in neuroscience —from the linear dimensionality reduction techniques such as Principal Component Analysis (PCA) Maćkiewicz & Ratajczak (1993)—to the non-linear visualization methods like Uniform Manifold Approximation and Projection (UMAP) McInnes et al. (2018) and t-Distributed Stochastic Neighbor Embedding (t-SNE) Kobak & Berens (2019)—and most recently, to the advanced, data-driven deep learning strategies. The focus was first on reducing the dimensionality of neuronal data alone. It later expanded to include joint dimensionality reduction with behavioral information and external stimuli (e.g., pi-VAE Prakash (2024) and BLEND Guo et al. (2025)). CEBRA represents a culmination of these advancements, integrating various techniques into a unified framework. CEBRA is used in this study to integrate the stimuli information experienced by each neuron as input for . NeuPRINT is a method used to extract invariant information from neurons Mi et al. (2023); however, it still adheres to traditional neuron modeling approaches, which can only implicitly represent neurons. This results in challenging training processes and suboptimal representation performance.

**Contrastive Learning for Voice Representations:** The basic approach of the solution presented in this paper draws inspiration from similar tasks, such as extracting inherent representations of speakers from voice data Torres et al. (2024). This type of work has been implemented on song-singer datasets, where contrastive learning is used to bring together the voices of the same singer while pushing apart the voices of different singers. However, while we borrowed the underlying idea, the specific methods had to be uniquely designed to accommodate the characteristics of neural data.

## 3 Method

### 3.1 Goal

Our goal is to develop a method for learning intrinsic neuron representations from neuron population data. These representations should meet three key criteria: (i) neurons with similar functional roles should exhibit greater similarity in their representations compared to those with different roles; (ii) the learned representations should be robust to variations in neuronal activity patterns caused by different stimuli or environmental conditions; and (iii) the representations should be adaptable and generalizable to new and unseen neuronal activity patterns.

### 3.2 Architechture

The ideal embedding space for neuron representations should cluster recording segments of the same neuron while also ensuring semantic consistency by placing similar neurons close to each other within the space. In line with the criteria outlined in Section 3.1, We conducted advanced contrastive learning loss functions, VICReg Bardes et al. (2021). We also carefully designed the data sampling methods to generate multi-segments activity data for each neuron for training purposes.

**Data sampling**: Without loss of generality, we take bi-segment data as an example. Consider two scenarios: 1) If a neuron has recordings from multiple sessions, we randomly extract two segments $(X_{seg}, X'_{seg})$ from different sessions. 2) If a neuron has recordings from only a single session, we randomly extract two non-overlapping segments $(X_{seg}, X'_{seg})$ from that session. We repeat this process $B$ times for a batch size of $B$, obtaining a positive pair batch $(X_{seg}^{(1)}, X_{seg}^{(2)})$.

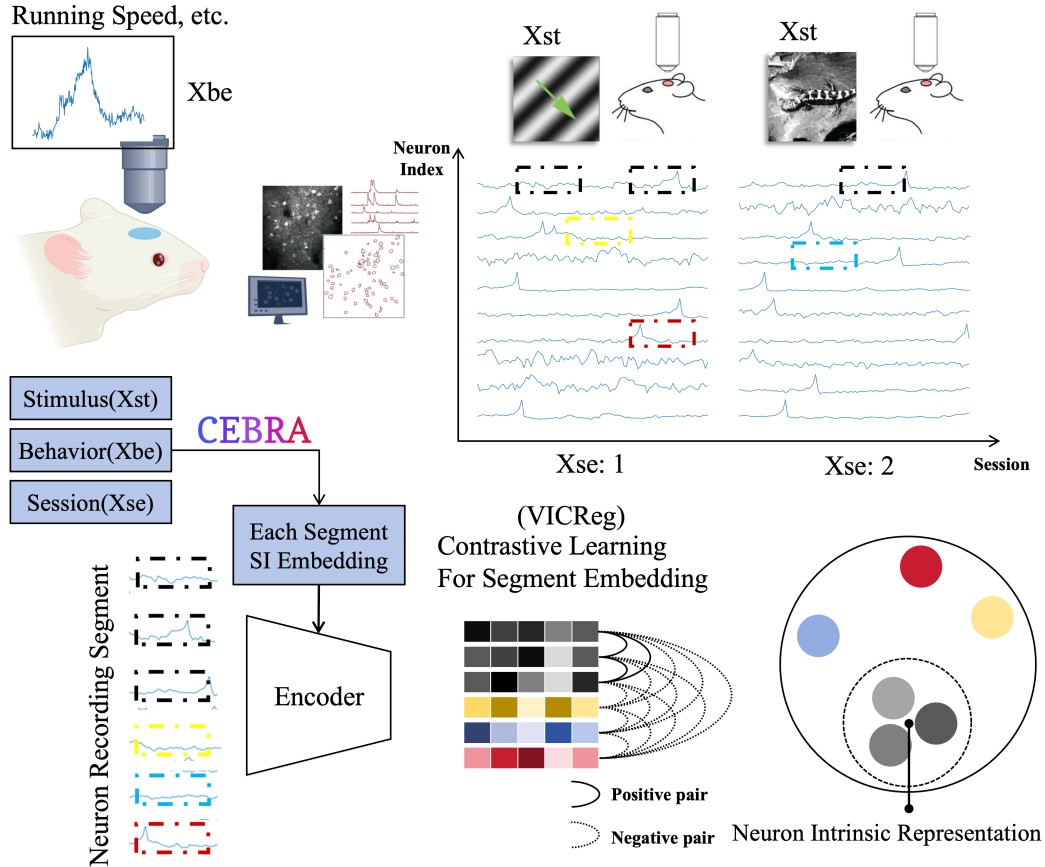

Figure 1: Optimization objective for obtaining intrinsic representations of neurons as follows: clips(segments) from the same neuron should have a higher average similarity than clips from different neurons. Figure 1 illustrates how this objective can be achieved using a contrastive learning approach, where different recording segments from the same neuron are treated as positive pairs and segments from different neurons are treated as negative pairs. Each neuron is considered separately, and the Stimulus, Behavior and Session treated as surrounding information(SI), which acts as auxiliary variables. The surrounding information for each neuron is processed and encoded using CEBRA.

**Model**: We should integrate surrounding information into each segment. In neurobiology, the stimulation received by the neuron corresponding to a segment ($X_{st}$), the animal's behavior at the corresponding time ($X_{be}$), and the session information ($X_{se}$) are all the surrounding information (SI) of this segment. CEBRA is a self-supervised learning algorithm designed to obtain interpretable and consistent embeddings of high-dimensional recordings using auxiliary variables. In simple terms, it serves as an encoder for the dynamic activity information of neuronal populations. Its advantage lies in its ability to encode both neuronal activity data (N*T) and corresponding auxiliary variables, such as behavior and external stimuli (M*T), into a lower-dimensional representation (D*T), where D represents the latent space dimensions. In CEBRA, D does not correspond to individual neurons, but rather provides a dimensionality reduction method that combines high-dimensional temporal activity data of neuronal populations with corresponding auxiliary variable information, aiming to uncover the relationships between neuronal activity and these variables.

Our work focuses on obtaining time-invariant representations for individual neurons. We innovatively use CEBRA as a preprocessing step for our input data from a new perspective. Specifically, from the viewpoint of individual neurons, we leverage CEBRA to integrate peripheral information—such as the activity of neighboring neuronal populations and behavioral data—associated with

each segment of a single neuron. $X_{st}$ consists of two components: peripheral neuronal activity and visual stimulation. The dimensions of $X_{st}$ are $n + 1$: The $n$ dimensions are the activity of the surrounding neurons relative to the neuron you're considering, because there will be coordinates for each neuron in the dataset, so for each neuron, we'll take the 47 nearest neurons in the experiment. The last dimension is visual stimuli, and we have a relatively simple way of dealing with visual stimuli: Blank visual stimuli are represented by 0, Drifting Grating visual stimuli by 1, Natural Scenes visual stimuli by 2, and each time point is represented by a fixed number. In the future, we could try to encode the video of visual stimuli so that it changes over time, but we currently only encode it according to the type of visual stimulus. The dimension of $X_{be}$ is 1: running speed each time point. The dimension of $X_{se}$ is 1: session number. The dimensions of $X_{si}$ are 10: The total dimension of $X_{be}$, $X_{se}$ and $X_{st}$ is 50, we use CEBBRA project them to the low dimension 10.

$$X_{si} = \text{CEBRA}(X_{st}, X_{be}, X_{se}) \tag{1}$$

We then denote the pair $(X_{si}, X_{seg})$ for the same time as $X$ in the following sections. The encoder $G(\cdot)$ maps the extracted input $X$ into a latent representation $H'$, which is then aggregated into time-invariant feature embeddings $H$ using adaptive average pooling. These embeddings $H$ are further mapped into a lower-dimensional space $Z$ by a projection layer $P(\cdot)$. The full model $F(\cdot)$ combines feature extraction, encoding, and projection. During training, $F(X)$ produces the projections $Z$. After training, the projection layer is removed, retaining only the embeddings $H$. The similarity between embeddings is measured using cosine similarity.

**Contrastive Learning - VICReg**: VICReg aims to enhance the quality of learned embeddings by incorporating three types of losses: variance loss, invariance loss, and covariance loss.

The invariance loss, ensures that embeddings of segments from the same neuron are close:

$$\mathcal{L}_{\text{invar}}(Z^{(1)}, Z^{(2)}) = \frac{1}{N} \sum_i \left\| Z_i^{(1)} - Z_i^{(2)} \right\|^2, \tag{2}$$

The variance loss regularizes the standard deviation of the embeddings to be near a target value $\mu$, which helps prevent embedding dimensions from becoming non-informative. Given $\mathbf{d_j}(\mathbf{z}) \in \mathbb{R}^B$, a vector of batch values at dimension $j$, the variance loss is defined as:

$$\mathcal{L}_{\text{var}}(\mathbf{z}) = \frac{1}{D} \sum_{j=1}^{D} \max\left(0, \mu - S\left(\mathbf{d_j}(\mathbf{z}), \epsilon\right)\right), \tag{3}$$

where $D$ represents the number of dimensions in $\mathbf{z}$, and $S$ denotes the regularized standard deviation, $S(x, \epsilon) = \sqrt{\text{Var}(x) + \epsilon}$.

The covariance loss promotes orthogonality among embedding dimensions by decorrelating them:

$$\mathcal{L}_{\text{cov}}(\mathbf{z}) = \frac{1}{D_z} \sum_{i \neq j} (C(\mathbf{z}))_{i,j}^2, \tag{4}$$

where $C(\mathbf{z}) = \frac{1}{N-1} \sum_{i=1}^{N} (\mathbf{z}_i - \bar{\mathbf{z}}) (\mathbf{z}_i - \bar{\mathbf{z}})^T$ is the covariance matrix of $\mathbf{z}$, and $\bar{\mathbf{z}} = \frac{1}{N} \sum_{i=1}^{N} \mathbf{z}_i$.

## 4 EXPERIMENTS

### 4.1 DATA

**Simulated Data:** Since the neuron intrinsic property is hardly available in vivo neuronal recordings, we applied to synthetic data where we can access the ground-truth intrinsic property. To make the synthetic data exhibit dynamics similar to that of real neurons, we simulated the data following the Izhikevich model. The Izhikevich model is a spiking neuron model that combines biological realism with computational efficiency. It is designed to capture the rich dynamics of real neurons while remaining computationally simple. The model is defined by the following differential equations:

$$\frac{dV}{dt} = 0.04V^2 + 5V + 140 - u + I, \tag{5}$$

$$\frac{du}{dt} = a(bV - u), \tag{6}$$

where $V$ is the membrane potential of the neuron, $u$ is a recovery variable, $I$ is the input current, and $a$, $b$, and $c$ are hyperparameters. The model also includes a spike-reset mechanism:

$$\text{if } V \geq 30 \text{ mV, then } \begin{cases} V \leftarrow c \\ u \leftarrow u + d \end{cases} \tag{7}$$

$a$, $b$, $c$ and $d$ can be regarded as intrinsic properties of each neuron.

**Real Data - Bugeon:** We utilized a rare, real-world multimodal dataset Bugeon et al. (2022) to test , which comprises two main components: 1) in vivo long-term recordings of multiple neurons in the mouse primary visual cortex using two-photon imaging technology, and 2) spatial transcriptomics of ex vivo slices from the recorded brain tissue to measure the expression levels of 72 selected genes in these neurons. Based on gene expression levels, neurons were assigned two main types of labels: (i) excitatory and inhibitory classification, and (ii) subclass labels (Lamp5, Pvalb, Vip, Sncg, and Sst) for some inhibitory neurons. This dataset contains recordings from four mice (Mouse A, B, C, and D). Mouse A has data from 6 sessions, Mouse B from 3 sessions, Mouse C from 5 sessions, and Mouse D from 3 sessions. During each session, the mice were exposed to three types of visual stimuli: Blank, Drifting Gratings, and Natural Scenes. A total of 9,278 unique neurons were recorded in the dataset.

**Real Data - Steinmetz:** The Steinmetz dataset comprises 39 Neuropixels recordings, capturing data from 400 to 700 neurons across various regions of the mouse brain of 10 mice during a visual behavior task Steinmetz et al. (2019). This dataset is excellent for exploratory analyses and is well-supported by extensive tutorial resources [1], alongside a wealth of experimental and behavioral variables included within.

## 4.2 EVALUATION

**Evaluation on Simulated Data:** The process is divided into two steps: (i) perform self-supervised contrastive learning on the dynamic data of all neurons to obtain a representation for each neuron; (ii) Employ a 5-fold cross-validation approach to use these neuron representations as input to a classifier for predicting the pre-defined neuron type labels from the simulation.

**Evaluation on Real Data - Bugeon:** The process is divided into three steps: (i) perform self-supervised contrastive learning on the dynamic data of all neurons from four mouse in the real dataset to obtain a representation for each neuron; (ii) Based on neurobiological prior knowledge and spatial transcriptomic gene expression information, assign cell type labels to each neuron. The labels fall into two categories: (a) excitatory and inhibitory, and (b) Lamp5, Pvalb, Vip, Sncg, and Sst; (iii) Implement a 4-fold (with the folds based on the identity of the mice) cross-validation approach where the neuron representations are used as input to a classifier to predict the neuron's type labels for both categories (a) and (b).

**Out-of-Domain Evaluation - Steinmetz dataset:** The process is divided into three steps: (i) Perform self-supervised contrastive learning on the dynamic data of all neurons from all mice in the real dataset to obtain a representation for each neuron; (ii) As before, assign location within brain regions labels to the neurons; (iii) Implement a 10-fold (with the folds based on the identity of the mice) cross-validation approach where the neuron representations are used as input to a classifier to predict the neuron's location. During neurodevelopment, where the position of a neuron is crucial for its differentiation, maturation, and connectivity. The location can influence the neuron's gene expression, synaptic connections, and ultimately its function Patel & Poo (1982). In this sense, the location is an intrinsic property because it defines role within the nervous system.

## 4.3 COMPARISON OF METHODS

**LOLCAT:** This method Schneider et al. (2023a) follows a supervised learning paradigm. It directly uses activity data from a subset of neurons to train a classifier to predict neuron labels, and then validates on the remaining neurons. Consequently, the representations learned in the intermediate

---

[1]https://www.youtube.com/watch?v=WXn4-FpVaOo

layers of the model contain only label information and do not fully capture the intrinsic properties of the neurons.

**PCA:** This method employs a self-supervised approach. Principal Component Analysis (PCA) reduces the dimensionality of each neuron's activity data by projecting it onto a lower-dimensional space, thereby providing a representation based on the most significant components of the activity data.

**UMAP:** Similar to PCA, Uniform Manifold Approximation and Projection (UMAP) is a self-supervised method that reduces the dimensionality of each neuron's activity data. UMAP preserves local and global structures in the data to create a meaningful lower-dimensional representation.

**NeurPrint:** This self-supervised method involves implicit representation learning through back-propagation. Due to its implicit nature, the model can be difficult to converge and may require substantial training data and time to achieve effective results, and the representations may not align well with the Platonic Representation Hypothesis.

To demonstrate that the representations learned only from neuronal activity capture invariant properties, we designed a verification process using simulation data. Based on the Izhikevich model, we defined the neurons' time-invariant hyperparameters as their intrinsic properties and applied stimulation to generate activity data (Figure 2). Our model relies solely on activity data to derive time-invariant representations of neurons and validates these representations by showing their ability to distinguish predefined intrinsic hyperparameters. To evaluate the effectiveness of our method, we classify intrinsic hyperparameters (five categories representing five neuronal firing modes) using the obtained intrinsic representations and compare its performance with four other algorithms: PCA, UMAP, NeuPRINT, and LOLCAT. The classification performance is assessed across multiple neuron types using three common metrics: Precision, Recall, and F1-score. Higher values of these metrics indicate better classification performance.

From the results presented in Table 4.3, we observe that: (i) our method consistently achieves the highest performance across most neuron firing modes, particularly in the Regular Spiking (RS) and Fast Spiking (FS) categories, where it significantly outperforms all baselines with F1-scores of 0.884 and 0.881, respectively; (ii) although some baselines, such as LOLCAT and NeuPRINT, demonstrate competitive results in certain neuron firing modes like Low-Threshold Spiking (LTS), their performance falls short compared to NeurPIR. For instance, in the case of Chattering (CH) neurons, NeurPIR achieves an F1-score of 0.671, surpassing LOLCAT's 0.652 and NeuPRINT's 0.611. These results underscore the robustness and accuracy of NeurPIR across varying neuron firing modes, establishing it as the most effective method in this comparison. Furthermore, these neuron firing modes are determined by preset hyperparameters, further validating that the intrinsic representations learned by NeurPIR effectively capture hyperparameter information.

### 4.4 REAL DATA - NEURON PLATONIC INTRINSIC REPRESENTATION CONTAINS MOLECULAR INFORMATION

In this section, we utilized a public multimodal dataset, referred to as Bugeon, to train and evaluate our model. We compared the performance metrics of neuron type classification for three methodologies: NeuPRINT, LOLCAT, and the proposed NeurPIR (PCA and UMAP were excluded as they cannot process behavioral information). The evaluation metrics included precision (Prec.), recall (Rec.), and F1 score, which are essential for assessing the classification effectiveness across various neuron types, categorized into subclasses and classes.

The results for the subclasses Lamp5, Vip, Pvalb, and Sst are summarized in Table A.1. For the Lamp5 subclass, NeurPIR achieved the highest F1 score (0.569 ± 0.014), demonstrating a balanced classification performance. Similarly, in the Vip subclass, NeurPIR outperformed others with an F1 score of 0.662 ± 0.035, highlighting its superior accuracy in classifying Vip neurons. For Pvalb neurons, NeurPIR again led with an F1 score of 0.604 ± 0.043, reflecting its effectiveness in this subclass. In the Sst subclass, although NeuPRINT achieved the highest precision (0.704 ± 0.168), its large standard deviation indicates variability. NeurPIR showed the highest F1 score (0.492 ± 0.080), demonstrating robustness despite the challenges in classifying Sst neurons.

Overall, the results demonstrate the effectiveness of NeurPIR in accurately classifying distinct neuronal subtypes, achieving improved precision and F1 scores compared to existing methodologies.

| Firing Modes | Metric | PCA | UMAP | NeuPRINT | LOLCAT | NeurPIR |
|---|---|---|---|---|---|---|
| regular spiking (RS) | Prec. | 0.689 | 0.473 | 0.836 | 0.783 | **0.872** |
| | Rec. | **0.918** | 0.590 | 0.908 | 0.910 | 0.898 |
| | F1. | 0.786 | 0.525 | 0.870 | 0.841 | **0.884** |
| intrinsically bursting (IB) | Prec. | 0.534 | 0.332 | 0.646 | **0.681** | 0.678 |
| | Rec. | 0.375 | 0.310 | 0.645 | 0.573 | **0.693** |
| | F1. | 0.440 | 0.320 | 0.644 | 0.620 | **0.684** |
| chattering (CH) | Prec. | 0.506 | 0.335 | 0.648 | 0.616 | **0.722** |
| | Rec. | 0.603 | 0.248 | 0.580 | **0.698** | 0.630 |
| | F1. | 0.548 | 0.285 | 0.611 | 0.652 | **0.671** |
| fast spiking (FS) | Prec. | 0.778 | 0.602 | 0.826 | 0.853 | **0.853** |
| | Rec. | 0.620 | 0.638 | 0.820 | 0.733 | **0.913** |
| | F1. | 0.689 | 0.618 | 0.823 | 0.787 | **0.881** |
| low-threshold spiking (LTS) | Prec. | 0.970 | 0.944 | 0.968 | **0.993** | 0.990 |
| | Rec. | 0.935 | 0.965 | **0.993** | 0.983 | 0.990 |
| | F1. | 0.952 | 0.954 | 0.980 | 0.988 | **0.990** |

Table 1: This table presents the performance metrics—precision (Prec.), recall (Rec.), and F1 score—across five methods (PCA, UMAP, NeurPrint, LOLCAT, and NeurPIR) for different neuron types: regular spiking (RS), intrinsically bursting (IB), chattering (CH), fast spiking (FS), and low-threshold spiking (LTS). The results indicate that and NeurPrint consistently achieve higher precision and F1 scores for most neuron types, while UMAP shows relatively lower performance, particularly for IB and CH neurons.

These findings highlight that the neuronal representations generated by our method effectively encode neuron type information. Since neuron types are determined by molecular characteristics, this suggests that the intrinsic representations also capture molecular information.

| Neuron Type | Metric | LOLCAT | NeuPRINT | NeurPIR |
|---|---|---|---|---|
| Subclass | Prec. | $0.460 \pm 0.064$ | $0.590 \pm 0.073$ | $0.607 \pm 0.049$ |
| | Rec. | $0.418 \pm 0.084$ | $0.562 \pm 0.093$ | $0.587 \pm 0.067$ |
| | F1. | $0.401 \pm 0.056$ | $0.553 \pm 0.080$ | $0.582 \pm 0.059$ |
| Class | Prec. | $0.619 \pm 0.015$ | $0.711 \pm 0.044$ | $0.755 \pm 0.023$ |
| | Rec. | $0.584 \pm 0.018$ | $0.707 \pm 0.047$ | $0.749 \pm 0.040$ |
| | F1. | $0.552 \pm 0.018$ | $0.706 \pm 0.047$ | $0.747 \pm 0.016$ |

Table 2: Average performance metrics for neuron type classification. Subclass averages are calculated across Lamp5, Vip, Pvalb, and Sst, while Class averages are calculated across Ex and In. Metrics include precision (Prec.), recall (Rec.), and F1 score, with errors represented as standard deviations.

| Method | Domain | Precision | Recall | F1 |
|---|---|---|---|---|
| LOLCAT | In-Domain | $0.701 \pm 0.093$ | $0.674 \pm 0.084$ | $0.694 \pm 0.072$ |
| | Out-of-Domain | $0.659 \pm 0.075$ | $0.644 \pm 0.092$ | $0.652 \pm 0.073$ |
| NeuPRINT | In-Domain | $0.764 \pm 0.091$ | $0.744 \pm 0.081$ | $0.743 \pm 0.071$ |
| | Out-of-Domain | $0.665 \pm 0.079$ | $0.678 \pm 0.092$ | $0.662 \pm 0.072$ |
| NeurPIR | In-Domain | $0.764 \pm 0.093$ | $0.749 \pm 0.072$ | $0.758 \pm 0.066$ |
| | Out-of-Domain | $0.701 \pm 0.081$ | $0.672 \pm 0.074$ | $0.708 \pm 0.054$ |

Table 3: Average performance metrics for In-Domain and Out-of-Domain data across all regions (Vis, Thal, Hipp, Mid) for each method. Metrics include precision, recall, and F1 score, with errors represented as standard deviations.

## 4.5 REAL DATA - SHOWS ROBUSTNESS ON OUT-OF-DOMAIN (UNSEEN ANIMAL) DATA

In this section, we focus on validating the consistency of the intrinsic representations generated by the model across different animals. The Steinmetz dataset, which includes neural activity data from ten rats, provides neuron labels based on their respective brain regions. It can be intuitively seen from Figure 3 that the response patterns of neurons in the same brain area are significantly different in different mice, but we hope that the representation obtained from the model still contains consistent information, like consistent brain area information. This corresponds to the evaluation of the generalizability of the representations obtained by the model on cross-modal (here, cross-animal) data in deep learning. We use a task of classifying location within brain regions to validate the intrinsic representations. We use NeurPIR to perform self-supervised training on all the neurons from the all mice to obtain the intrinsic representations. For the downstream task of brain region classification, we used 10-fold cross-validation (with folds based on the identity of the mice).

As shown in Figure 4.4, the model's generalizability is evaluated through brain region classification across different neural representations. In-domain validation results demonstrate that consistently outperforms other methods like LOLCAT and NeuPRINT across all location within brain regions, including visual cortex (ViS), thalamus (Thal), hippocampus (Hipp), and midbrain (Mid). Specifically, achieves validation Precision close to 0.80 in most regions, with NeuPRINT slightly trailing behind.

When examining out-of-domain performance (cross-animal), we notice a general drop in accuracy for all methods. However, NeurPIR still retains a higher degree of accuracy across all location within brain regions compared to NeuPRINT and LOLCAT, showing the model's ability to capture more robust intrinsic representations across different animals. This consistent outperformance across both in-domain and out-of-domain tests suggests that the representations obtained by better generalize across animals while still preserving critical brain region information.

## 5 CONCLUSION AND DISCUSSION

In this paper, we present a novel and scalable approach for extracting and leveraging the intrinsic properties of neurons. This approach holds significant potential for re-evaluating existing neuroscience data and enhancing our understanding of neural computation. Future research may focus on further enhancing its ability to handle an even more diverse range of datasets and applying it to other domains where extracting intrinsic properties from complex systems is of great importance. Limitations: (i) The representation learned by our method can only distinguish neurons with substantial differences in essential attributes. For instance, when neurons are associated with more refined brain-area labels, it becomes challenging to differentiate them, necessitating more data for training support. (ii) The learned neuron representations only support data collected from the same technology. The generalization of cross - platform data, such as two-photon data and neurpixel data, remains to be explored. (iii) Over very long timescales, some of the short-term invariant properties of neurons may change. This can be exploited to study the changes in neuronal properties during the progression of diseases like Alzheimer's disease.

## REPRODUCIBILITY STATEMENT

To enhance the reproducibility of this study, we provide an Appendix section comprising 4 subsections that offer detailed supplementary information. Appendix A.3 presents the pseudo-code of Synthetic Data. Appendix A.4 presents python code for downloading and organizing the steinmetz dataset. Appendix A.5 presents the pseudo-code of sownstream task. Appendix A.6 presents Description and Function across firing types / neuron types / brain regions in this paper.

## ACKNOWLEDGEMENT

This research was supported by National Natural Science Foundation of China (6220071694), National Key Research and Development Program of China (2024YFF0507404), and Young Elite Scientist Sponsorship Program by Beijing Association for Science and Technology (BYESS2023026).

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

## A APPENDIX

A.1  DETAIL PERFORMANCE METRICS FOR NEURON TYPE/LOCATION CLASSIFICATION.

| Neuron Type | Metric | NeuPRINT | LOLCAT | NeurPIR |
|---|---|---|---|---|
| Subclass | | | | |
| Lamp5 | Prec. | 0.481 ± 0.035 | 0.344 ± 0.023 | 0.487 ± 0.007 |
| | Rec. | 0.667 ± 0.037 | 0.694 ± 0.033 | 0.684 ± 0.028 |
| | F1. | 0.559 ± 0.036 | 0.460 ± 0.024 | 0.569 ± 0.014 |
| Vip | Prec. | 0.614 ± 0.044 | 0.592 ± 0.078 | 0.657 ± 0.048 |
| | Rec. | 0.652 ± 0.034 | 0.406 ± 0.028 | 0.668 ± 0.027 |
| | F1. | 0.632 ± 0.034 | 0.480 ± 0.042 | 0.662 ± 0.035 |
| Pvalb | Prec. | 0.559 ± 0.046 | 0.428 ± 0.026 | 0.602 ± 0.042 |
| | Rec. | 0.604 ± 0.017 | 0.403 ± 0.017 | 0.607 ± 0.055 |
| | F1. | 0.580 ± 0.032 | 0.415 ± 0.019 | 0.604 ± 0.043 |
| Sst | Prec. | 0.704 ± 0.168 | 0.477 ± 0.130 | 0.681 ± 0.072 |
| | Rec. | 0.323 ± 0.084 | 0.170 ± 0.077 | 0.388 ± 0.080 |
| | F1. | 0.441 ± 0.105 | 0.248 ± 0.100 | 0.492 ± 0.080 |
| Class | | | | |
| Ex | Prec. | 0.685 ± 0.050 | 0.555 ± 0.010 | 0.720 ± 0.025 |
| | Rec. | 0.774 ± 0.020 | 0.854 ± 0.009 | 0.817 ± 0.031 |
| | F1. | 0.726 ± 0.036 | 0.673 ± 0.008 | 0.765 ± 0.009 |
| In | Prec. | 0.737 ± 0.037 | 0.682 ± 0.020 | 0.790 ± 0.020 |
| | Rec. | 0.640 ± 0.073 | 0.314 ± 0.027 | 0.680 ± 0.048 |
| | F1. | 0.685 ± 0.058 | 0.430 ± 0.028 | 0.729 ± 0.023 |

Table 4: Performance metrics for neuron type classification, 4-fold cross-validation was used, with the folds based on the identity of the mice This table delineates the precision (Prec.), recall (Rec.), and F1 score for various neuron types categorized into subclasses (Lamp5, Vip, Pvalb, Sst) and classes (Ex, In). The metrics demonstrate the comparative performance of these methods in identifying and classifying distinct neuronal subtypes.

| Region | Method | Precision | Recall | F1 |
|---|---|---|---|---|
| Vis | neu | 0.923 ± 0.029 | 0.614 ± 0.052 | 0.736 ± 0.042 |
| | | 0.751 ± 0.047 | 0.467 ± 0.105 | 0.572 ± 0.086 |
| | LOLCAT | 0.836 ± 0.049 | 0.450 ± 0.059 | 0.584 ± 0.060 |
| | | 0.775 ± 0.014 | 0.425 ± 0.053 | 0.547 ± 0.043 |
| | NeurPIR | 0.900 ± 0.016 | 0.652 ± 0.024 | 0.756 ± 0.013 |
| | | 0.825 ± 0.017 | 0.514 ± 0.012 | 0.633 ± 0.013 |
| Thal | NeuPRINT | 0.758 ± 0.025 | 0.762 ± 0.005 | 0.760 ± 0.013 |
| | | 0.669 ± 0.025 | 0.724 ± 0.019 | 0.695 ± 0.016 |
| | LOLCAT | 0.717 ± 0.047 | 0.738 ± 0.013 | 0.726 ± 0.025 |
| | | 0.698 ± 0.073 | 0.681 ± 0.010 | 0.688 ± 0.040 |
| | NeurPIR | 0.765 ± 0.025 | 0.772 ± 0.018 | 0.768 ± 0.015 |
| | | 0.725 ± 0.049 | 0.761 ± 0.027 | 0.742 ± 0.034 |
| Hipp | NeuPRINT | 0.708 ± 0.033 | 0.785 ± 0.013 | 0.744 ± 0.023 |
| | | 0.626 ± 0.026 | 0.671 ± 0.022 | 0.647 ± 0.016 |
| | LOLCAT | 0.668 ± 0.034 | 0.742 ± 0.018 | 0.703 ± 0.027 |
| | | 0.595 ± 0.028 | 0.703 ± 0.026 | 0.644 ± 0.020 |
| | NeurPIR | 0.723 ± 0.018 | 0.782 ± 0.019 | 0.751 ± 0.013 |
| | | 0.644 ± 0.032 | 0.743 ± 0.019 | 0.689 ± 0.017 |
| Mid | NeuPRINT | 0.669 ± 0.026 | 0.814 ± 0.005 | 0.734 ± 0.017 |
| | | 0.611 ± 0.012 | 0.748 ± 0.031 | 0.672 ± 0.013 |
| | LOLCAT | 0.584 ± 0.018 | 0.768 ± 0.010 | 0.663 ± 0.013 |
| | | 0.569 ± 0.024 | 0.726 ± 0.029 | 0.637 ± 0.009 |
| | NeurPIR | 0.669 ± 0.028 | 0.789 ± 0.018 | 0.724 ± 0.021 |
| | | 0.634 ± 0.027 | 0.740 ± 0.013 | 0.683 ± 0.021 |

Table 5: Performance Comparison of Methods Across location within brain regions (In-Distribution vs Out-of-Distribution). The upper part of each indicator represents In Domain and the lower part represents Out of Domain.

## A.2 HYPERPARAMETERS OF FIVE FIRING MODES OF NEURONS WHEN SIMULATING DATA

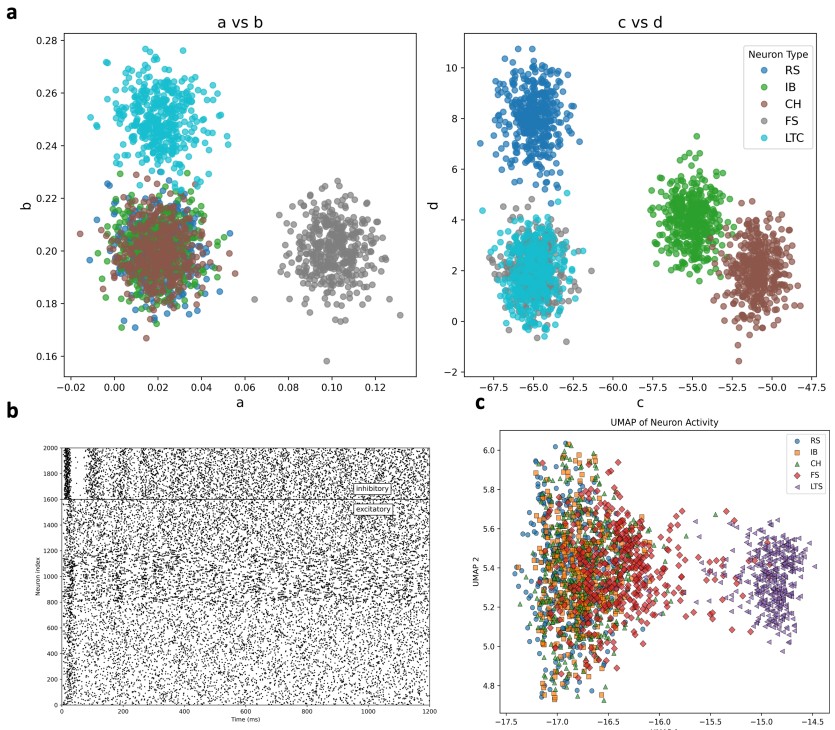

Figure 2: a: the hyperparameters of five firing modes of neurons when simulating data; b: the neuron firing: the index of regular spiking (RS) is 0-400, the index of intrinsically bursting (IB) is 400-800, the index of chattering (CH) is 800-1200, the index of fast spiking (FS) is 1200-1600, the index of low-threshold spiking (LTS) is 1600-2000; c: the results of the visualization of neuronal activity data using UMAP.

### A.3 Simulation of Neuronal Population Using the Izhikevich Model

1. Define the Izhikevich model parameters for each neuron type:
    - Regular Spiking (RS)
    - Intrinsically Bursting (IB)
    - Chattering (CH)
    - Fast Spiking (FS)
    - Low-Threshold Spiking (LTS)
2. Initialize the total number of excitatory ($N_e$) and inhibitory ($N_i$) neurons.
3. Assign neuron types to indices:
    - 25% Regular Spiking (RS)
    - 25% Intrinsically Bursting (IB)
    - 25% Chattering (CH)
    - 25% Fast Spiking (FS)
4. Initialize synaptic connection matrix $S$.
5. Set initial values for membrane potential $v$ and recovery variable $u$.
6. For each time step $t$ from 0 to $T$:
    - Calculate input current $I$.
    - If $t > 0$, add synaptic contributions from previously fired neurons.
    - Update membrane potential $v$ and recovery variable $u$ using Euler's method.
    - Check for spikes and record firing events.
    - Reset membrane potential and increment recovery variable for fired neurons.

- Store current and potentials for analysis.

7. Update neuron activity data.

8. Plot firing events.

9. Print shapes of activity arrays for each neuron type.

## A.4 PYTHON CODE FOR DOWNLOADING AND ORGANIZING THE STEINMETZ DATASET

```python
# @title Data retrieval
import os, requests

fname = []
for j in range(3):
    fname.append('steinmetz_part%d.npz' % j)
url = ["https://osf.io/agvxh/download"]
url.append("https://osf.io/uv3mw/download")
url.append("https://osf.io/ehmw2/download")

for j in range(len(url)):
    if not os.path.isfile(fname[j]):
        try:
            r = requests.get(url[j])
        except requests.ConnectionError:
            print("!!! Failed to download data !!!")
        else:
            if r.status_code != requests.codes.ok:
                print("!!! Failed to download data !!!")
            else:
                with open(fname[j], "wb") as fid:
                    fid.write(r.content)
# @title Data loading
alldat = np.array([])
for j in range(len(fname)):
    alldat = np.hstack((alldat,
                        np.load('steinmetz_part%d.npz' % j,
                                allow_pickle=True)['dat']))
```

### A.5 PSEUDO-CODE OF DOWNSTREAM TASK

---

**Algorithm 1** Neural Network Classification with K-Fold Cross-Validation

---

1: **Import necessary libraries:** NumPy, scikit-learn, Seaborn, Matplotlib
2: **Ignore warnings**
3:
4: **procedure** MAIN
5:     **Initialize labels** as NumPy array
6:     **Encode labels** using LabelEncoder
7:     **Standardize neuron features** using StandardScaler
8:
9:     **Initialize StratifiedKFold** with 5 splits
10:     **Initialize statistics dictionaries** for precision, recall, F1-score
11:     **Initialize empty list for confusion matrices**
12:
13:     **for** each fold in K-Fold **do**
14:         **Split data into training and test sets**
15:         **Create MLPClassifier model**
16:         **Fit model to training data**
17:         **Predict labels for test data**
18:         **Generate classification report and confusion matrix**
19:         **Append confusion matrix to list**
20:
21:         **for** each cell type in classes **do**
22:             **Record precision, recall, and F1-score**
23:         **end for**
24:     **end for**
25:
26:     **for** each cell type in classes **do**
27:         **Calculate and print average metrics**
28:     **end for**
29:
30:     **Compute cumulative confusion matrix**
31:     **Print cumulative confusion matrix**
32:
33:     **Optional: Plot cumulative confusion matrix using Seaborn**
34: **end procedure**

---

## A.6 Steinmetz dataset

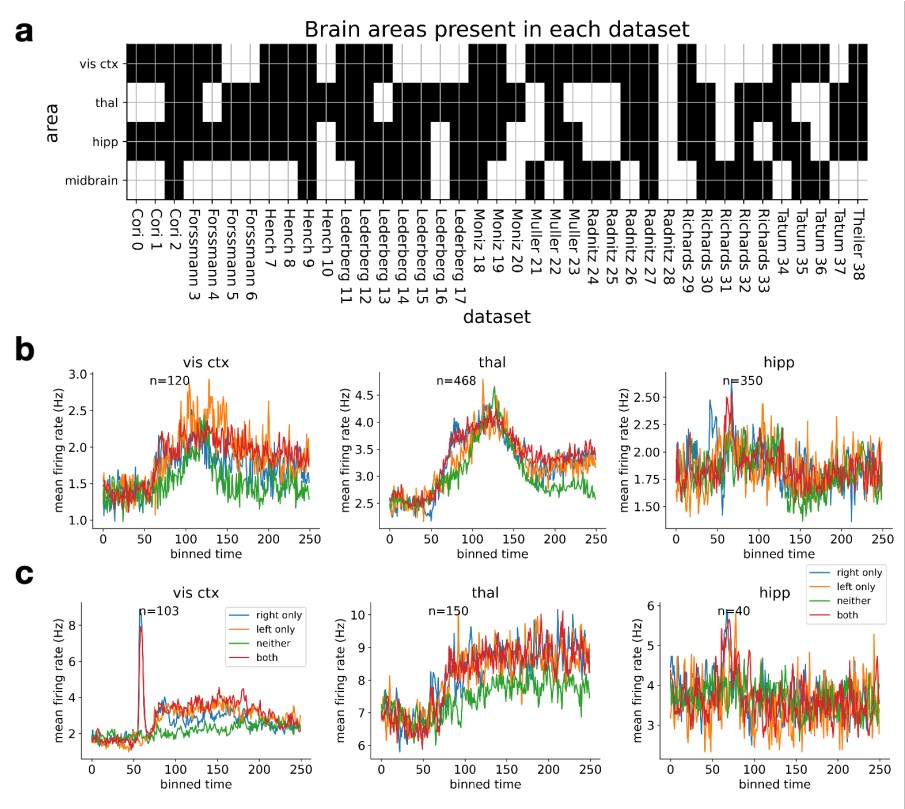

Figure 3: a: Steinmetz dataset contains 39 subdataset from 10 mice. Each subdataset records 400-700 neurons each from across the mouse brain during a visual behavior task. Each neuron with its brain area label. b: show of part of subdataset3. c: show of part of subdataset26.

## A.7 Description and Function across firing types / neuron types / brain regions in this paper

Firing Types:

1. Regular Spiking (RS) Neurons:

- Description: Regular Spiking neurons are characterized by their ability to fire action potentials at a regular, predictable rate in response to a sustained depolarizing stimulus. These neurons typically exhibit a linear relationship between input and output, meaning they can respond to small inputs with a consistent firing pattern.

2. Intrinsically Bursting (IB) Neurons:

- Description: Intrinsically Bursting neurons can produce bursts of action potentials in response to a depolarizing stimulus, in addition to regular single spikes. This type of neuron displays a unique pattern of activity where a series of action potentials is generated in quick succession followed by a period of quiescence.

3. Chattering (CH) Neurons:

- Description: Chattering neurons exhibit a high-frequency, sustained firing pattern. These neurons are characterized by their ability to fire at a high rate in bursts, often exhibiting a very rapid oscillatory behavior with minimal latency between spikes.

4. Fast Spiking (FS) Neurons:

- Description: Fast Spiking neurons are a type of inhibitory interneuron known for their ability to fire action potentials at very high frequencies (often greater than 100 Hz). They exhibit rapid and precise spiking in response to stimuli and are critical for the regulation of network activity.

5. Low-Threshold Spiking (LTS) Neurons:

- Description: Low-Threshold Spiking neurons are characterized by their ability to fire action potentials at relatively low levels of depolarization. They are often described as having a "sensitive" or "easy-to-trigger" firing threshold, which enables them to respond to subtle changes in membrane potential.

Neuron Types:

1. Lamp5 Neurons (Lysosomal-Associated Membrane Protein 5):

- Description: Lamp5 neurons are a type of GABAergic interneuron that expresses the Lamp5 protein, which is involved in cellular trafficking and autophagy. These neurons are often found in regions of the brain involved in cortical circuits, particularly in the cortex.

2. Vip Neurons (Vasoactive Intestinal Peptide-expressing neurons):

- Description: Vip neurons are a type of inhibitory interneuron that express vasoactive intestinal peptide (VIP), a neuropeptide involved in the modulation of neural circuits. These neurons typically have broad inhibitory effects in cortical regions and influence the activity of other interneuron types.

3. Pvalb Neurons (Parvalbumin-expressing neurons):

- Description: Pvalb neurons are a subtype of fast-spiking inhibitory interneurons that express the calcium-binding protein parvalbumin. These neurons are well known for their ability to fire action potentials at extremely high frequencies with minimal delay.

4. Sst Neurons (Somatostatin-expressing neurons):

- Description: Sst neurons are another type of inhibitory interneuron that express somatostatin, a neuropeptide that inhibits neurotransmitter release. These neurons are typically involved in modulating cortical circuits and are especially important for regulating synaptic plasticity.

Brain Regions:

1. Vis (Visual Cortex): Occipital lobe, primarily in the calcarine sulcus and surrounding regions.

2. Hipp (Hippocampus): Medial temporal lobe, beneath the cerebral cortex.

3. Thal (Thalamus): Deep within the brain, just above the brainstem, near the center.

4. Mid (Midbrain): Between the forebrain and hindbrain, just above the pons and below the thalamus.

