# OpenReview forum: "Neuron Platonic Intrinsic Representation From Dynamics Using Contrastive Learning"
_ICLR.cc/2025/Conference — ICLR 2025 Poster_

### Official Review · Reviewer_Yi7j · 2024-10-19

**Soundness:** 3
**Presentation:** 3
**Contribution:** 3
**Rating:** 8
**Confidence:** 4

**Summary:**

The paper proses a novel method called NeurPIR for extracting individual neuron representation based of neuron population recordings in a self-supervised fashion. For this purpose NeurPIR essentially combines a neural data specific sampling method, average pooled CEPRA embeddings, and a VICReg contrastive loss in a cohesive manner. The method is evaluated on a suite of synthetic and real neuron population activity recordings, where it superior performance to alternative methods.

**Strengths:**

- The paper tackles a relevant, yet extremely challanging task: extracting individual neuron characteristics from neuron population data.
- The paper convincingly demonstrates the proposed methods ability to do so to some extent, in particular compared to other available methods.
- The related literature, methods and experiments section is very detailed and well written, aiding interpretability of the results and reproducibility.

**Weaknesses:**

* 1\) The Steinmetz dataset experiment: likely doeant support the claim of being able to extract neuron internal represenations, as the method was trained to predict the location of the neurons, arguably an external neuron property?
   - Even if two exactly same neurons were integrated into two different brain regions they are likely identifiable through their activity as the differences in input to the two regions typically significantly deviates.
   - To claim exactraction of neuron intrinsic properties, the above effect would have to be significantly smaller than neuron intrinsic properly related differences, which is not validated.
* 2\) The Conclusion: is crucially missing a paragraph on the limitations of the proposed method and the experimental findings. The paper would significantly benefit from it.
* 3\) The Abstract: the first half is very confusing to read and doesn’t make it clear what the paper is actually about (for someone from a general computational neuroscience background). The paper would therefore greatly benefit from simpler and more concrete wording there.
   - Examples of terms that were not really helpful to me : “decoupling of intrinsic properties, “time-varying dynamics”, “dynamic activities”, “varying signals”, etc. What property, whos dynamics, what activity, which signal?
   - In particular “intrinsic properties” should be immediately followed by examples (later given), and the first mention of PRH feels out of place and its unclear how it relates to the sentences surrounding it.
   - Furthermore, in the context of computational neuroscience “what information is conveyed by neural activities” most often implies figuring out what neurons try to communicate to process information, which the paper is not about.
   - Finally, a statement like “NeurPIR captures the preset hyperparameters of each neuron” implies precise recovery e.g. $b = 0.2$. The proposed method was rather shown to capture rough categorical differences instead. More appropriate would could be e.g. “NeurPIR captures the class/category/type of each neuron”.
* 4\) The Figures: have barely readable label sizes and legends, or missing annotation.
   - Figures 2 and 3 feature barely readable label sizes and legend
  - Figure 1 could use more labels that help relate the model description to the images in the figure. (e.g. X, H, Z, P, F, CEBRA, VICReg)
* 5\) The Tables (or their descriptions): would benefit from some aggregate metrics across all categories.
* 6\) For Reproducibility: manual labeling like in the Bugeon is hard to reproduce, and its unclear to me from the text whether these will be / are provided.

**Questions:**

1) How exactly are the experiments on the Steinmetz dataset supporting the claims of neuron intrinsic representation learning?
2) For the Bugeon Dataset: why was only data of mice A used?
3) For the Steinmetz Dataset; why wan’t also e.g. a 10-fold crossvalidation used (folds along the mice identity)?
4) Would you expect you method to also perform well if the number of Izhikevich neuron “types” was greatly increased (e.g. 10,20,30 categories)?
5) Will/are the exact labels used for three experiments, and in particular for Bugeon be available for others to be able to reproduce the experiments exactly?
6) Would you expect different results if max-pooling was used instead of mean-pooling?
7) Is there any more literature / papers exactly doing single neuron characterization based on neuron population activity (possibly on other datasets)? It seems to be challenging to find more related literature.

---

> ### Author Response · Authors · 2024-11-14
>
> How exactly are the experiments on the Steinmetz dataset supporting the claims of neuron intrinsic representation learning?
>
> -The intrinsic representation of a neuron is, in fact, a time-invariant representation. We obtain a time-invariant representation for each neuron through a self-supervised approach. The question then arises: how can we validate the effectiveness of this time-invariant property? To address this, we take into account the time-invariant attributes of neurons that are already known from prior knowledge, such as neuron type and the brain region in which the neuron is located.
>
> For the Bugeon Dataset: why was only data of mice A used?
>
> -Thanks for the suggestion, the cross animal results are more meaningful for neuroscience, so we changed the evaluations from one animal to multiple animals. Specifically, we have four mice, so 4-fold was used, with the folds based on the identity of the mice. We also divided a validation set and used a random search strategy to tune the hyperparameters of methods being compared, employing the best models on the test set. We have revised the result in the paper.
>
> For the Steinmetz Dataset; why wan’t also e.g. a 10-fold crossvalidation used (folds along the mice identity)?
>
> -Following your suggestion, we have reconducted the experiments and used 10-fold cross-validation (with folds based on the identity of the mice). We also replaced the original bar charts with a table to present more detailed information, including the precision, recall, and F1 scores details.
>
> Would you expect you method to also perform well if the number of Izhikevich neuron “types” was greatly increased (e.g. 10,20,30 categories)?
>
> -Strictly speaking, the performance of the model depends on the similarity between different neuron types, rather than the number of types. If the added categories are very similar to each other, the model's performance is likely to decrease. However, if the added categories still exhibit distinguishable differences, the performance should remain unchanged.
>
> Will/are the exact labels used for three experiments, and in particular for Bugeon be available for others to be able to reproduce the experiments exactly?
>
> -Of course, I am willing to provide a jupyternotebook[1] to show how to preprocess Bugeon dataset[3], download dataset from [2].
>
> Would you expect different results if max-pooling was used instead of mean-pooling?
>
> -I believe that max pooling will not perform better than average pooling, because before the neuronal data is input, it has already undergone binning of the firing rate, which is already similar to a max pooling operation. If we were to use max pooling again, it would lead to a significant loss of information.
>
> Is there any more literature / papers exactly doing single neuron characterization based on neuron population activity (possibly on other datasets)? It seems to be challenging to find more related literature.
>
> -I came up with this idea when I saw that there's work now using deep learning to learn the inherent properties of different singers voices, and you can refer to the papers that inspired my idea[4].
>
> [1]https://drive.google.com/file/d/1Gf9RS49K2W0npLE3kY77GD--s0oHgXRA/view?usp=drive_link
> [2]https://figshare.com/articles/dataset/A_transcriptomic_axis_predicts_state_modulation_of_cortical_interneurons/19448531
> [3]https://www.nature.com/articles/s41586-022-04915-7
> [4]https://dl.acm.org/doi/10.1109/TASLP.2022.3169627

---

> > ### Comment · Reviewer_Yi7j · 2024-11-14
> > **Further clarifications needed, comment on novelty:**
> >
> > Thank you for the clarifications and improvements regarding questions 2)-7).
> >
> > Regarding question 1) the explanation is too general; could you please more concretely explain why the Steinmetz Dataset is suitable to learn “neuron internal representations” as opposed to “neuron external representations”?
> >
> > Furtheremore, could you please specify whether for the final manuscript you would address points 2)-4) of the listed weaknesses?
> >
> > Unlike reviewer **bVqe**, I do not see a problem with the novelty of the method or research question, even if it consists of the straightforward combination of well known computation blocks, and as mentioned, I am not aware of any papers investigating the same research question. Therefore I believe this is a very interesting contribution to the field, and a distinctive strength of the paper. I’m open to evidence to the contrary in terms of related literature.

---

> > > ### Author Response · Authors · 2024-11-14
> > >
> > > why the Steinmetz Dataset is suitable to learn “neuron internal representations” as opposed to “neuron external representations”?
> > >
> > > - You're right, the brain regions where neurons are located can indeed be expressed as external properties of neurons, but notice that the words we use are ’intrinsic‘, whether external or internal, as long as they are time-invariant, we can regard them as intrinsic for a neuron.
> > >
> > > 2)-4) of the listed weaknesses:
> > >
> > > - Limitations: (1) The representation learned by our method can only distinguish neurons with large differences in essential attributes. For example, if neurons consider more refined brain area labels, it is difficult to distinguish them, which requires more data to support training. (2) The learned neurons represent that only data collected from the same technology are supported, and the generalization of cross-platform data, such as two-photon data and neurpixel data, remains to be explored. (3) Considering very long timescales, it is possible that some of the short-term invariant properties of neurons may change, which can be used to study changes in neuronal properties during the development of diseases such as Alzheimer's disease.
> > > - Abstract (revised version): The Platonic Representation Hypothesis posits that behind different modalities of data (what we sense or detect), there exists a universal, modality-independent representation of reality. Inspired by this, we treat each neuron as a system, where we can detect the neuron’s multi-segment activity data under different peripheral conditions. We believe that, similar to the Platonic idea, there exists a time-invariant representation behind the different segments of the same neuron, which reflects the intrinsic properties of the neuron’s system. Intrinsic properties include the molecular profiles, brain regions and morphological structure, etc. The optimization objective for obtaining the intrinsic representation of neurons should satisfy two criteria: (I) segments from the same neuron should have a higher similarity than segments from different neurons; (II) the representations should generalize well to out-of-domain data. To achieve this, we employ contrastive learning, treating different segments from the same neuron as positive pairs and segments from different neurons as negative pairs. During the implementation, we chose the VICReg, which uses only positive pairs for optimization but indirectly separates dissimilar samples via regularization terms. To validate the efficacy of our method, we first applied it to simulated neuron population dynamics data generated using the Izhikevich model. We successfully confirmed that our approach captures the type of each neuron as defined by preset hyperparameters. We then applied our method to two real-world neuron dynamics datasets, including spatial transcriptomics-derived neuron type annotations and the brain regions where each neuron is located. The learned representations from our model not only predict neuron type and location but also show robustness when tested on out-of-domain data (unseen animals). This demonstrates the potential of our approach in advancing the understanding of neuronal systems and offers valuable insights for future neuroscience research.
> > > - I have redrawn the picture to make legend and caption clearer and increased it to a resolution of 600dpi.
> > > - I have resubmitted the revised manuscript.
> > >
> > > Thank you for your advice. I hope you can consider it again!

---

> ### Comment · Reviewer_Yi7j · 2024-11-14
> **I will adjust my scores accordingly, minor things remain.**
>
> Thank you once again for the clarifications and addressing the weaknesses. I will adjust my soundness score and overall score accordingly.
>
> The distinction between “intrinsic” vs “internal” is a valid point. However, then an explanation is required as to how an “external property” such as location of an object can be an “intrinsic property” of said object, which is not obvious and therefore **must** be included e.g. with the introduction of the dataset. I suppose that indeed for neurons the location of a specific neuron is not independent of its other properties and inherently linked (see developmental neurobiology). This would then explain the validity of the Steinmetz dataset experiments.
>
> Thank you for the revised abstract and included limitations. I will adjust my presentation score and overall score accordingly.
>
> 1) Regarding the abstract: it is **much** more understandable and straightforward to me. I especially appreciate addressing each previous point carefully. Some remaining ambiguities, and related potential suggestion e.g.:
> - “different segments of the same neuron” -> “different recording segment from the same neuron”? (line 16)
> - “brain regions” -> “location within brain regions”? (line 18)
> - now the method name NeuPIR or Neu**r**PIR is not mentioned until page 4 or 7 -> probably it should be included already in the abstract and it should definitely be used in a consistent manner; one **xor** the other.
>
> 2) Regarding Figures and Tables: I still believe Figure 1 could benefit from more labels such as X, H, Z, P, F or at least CEBRA, VICReg. Tables should not go over the margins (e.g. Table 3), and Titles of subplots should be appropriately sized (e.g. Figure 2). Consider also using PDF figures for quality improvement instead. Finally, aggregate metrics across firing types / neuron types / brain regions would still help to compare the methods at a glance (at least in appendix).
>
> I will assume in good faith that a Steinmetz dataset explanation will be included and the minor complaints (in so far the authors agree) will be addressed in the final versions of the manuscript. The new soundness and presentation score is 3 each, and the overall score an 8.

---

> > ### Author Response · Authors · 2024-11-14
> >
> > - I add "It is worth noting that the location of the brain region where the neuron is located is an external property, which itself cannot be regarded as a intrinsic property, but we assume that this external property is constant in the short experimental time range, indirectly as an intrinsic property of the neuron." to explain the validity of the Steinmetz dataset experiments.
> > Regarding the intrinsic properties of the outside of neurons, this is a topic worth continuing to explore. There may also be external properties that remain unchanged until death.
> >
> > - The details in the abstract have been modified as you suggested. (line 16 and 18)
> > - We use NeurPIR uniformly and introduce it in the introduction.
> > - We fix the table overflow issue.
> > - We add a description of firing types / neuron types / brain regions to the appendix.
> > - We add the note above picture 1.
> > Thank you for your advice. I hope you can consider it again!

---

> > > ### Author Response · Authors · 2024-11-15
> > >
> > > - I revised the explanation of the the validity of the Steinmetz dataset experiments again, and I referred to the point of view of developmental neurobiology: "During neurodevelopment, where the position of a neuron is crucial for its differentiation, maturation, and connectivity[1]. The location can influence the neuron's gene expression, synaptic connections, and ultimately its function. In this sense, the location is an intrinsic property because it defines role within the nervous system."
> > >
> > > [1]Patel N, Poo M M. Orientation of neurite growth by extracellular electric fields[J]. Journal of Neuroscience, 1982, 2(4): 483-496.
> > >
> > > My previous explanation misunderstood your meaning. Please check whether the above new explanation is reasonable.
> > > I have revised it in Line 292.

---

> > > > ### Comment · Reviewer_Yi7j · 2024-11-15
> > > > **Minor things again.**
> > > >
> > > > Thank you for the second revised Steinmetz dataset justification; overall it makes sense now. However, it has minor issues. See the following suggestion in this case:
> > > >
> > > > “During neurodevelopment, the position of a neuron is crucial for its differentiation, maturation, and connectivity[1]. The location can influence the neuron’s gene expression, synaptic connections, and ultimately its function Patel & Poo (1982). In this sense, the location is an intrinsic property of a neuron because it defines its role within the nervous system.”
> > > >
> > > > Furthermore, I’m afraid I was also not clear enough about my suggestion wrt aggregate metrics. E.g. for Table 01 as it is now, the results for each firing mode are listed separately. However, unless this is central to the point, one could also list average Precision, Recall, and F1-score where the average is calculated over all firing modes, to demonstrate each method's efficacy.
> > > >
> > > > Also please note the page limit for the final submission. Potentially one could display aggregate metrics in the main text where appropriate, and move the detailed tables to the appendix, thereby also addressing the page limit issue.

---

### Official Review · Reviewer_bVqe · 2024-10-30

**Soundness:** 3
**Presentation:** 3
**Contribution:** 2
**Rating:** 6
**Confidence:** 3

**Summary:**

This paper proposes a contrastive learning method (NeuPIR) for analyzing single-neuron activity data, with the goal of obtaining a representation which preserves property-level similarity of neurons (e.g. cell type). The method combines a variational autoencoder (CEBRA) with a contrastive loss (VICreg). NeuPIR is applied to a synthetic dataset and real neural datasets with cell type and brain area information, and compared against a few other methods of feature extraction (PCA, UMAP, NeuPRINT, & LOLCAT).

**Strengths:**

* Understanding the role of cell type diversity is a major challenge for neuroscience. A contrastive approach like this one, which can in principle be trained without labels, is likely to be the way forward given the difficulty of obtaining cell type information and activity simultaneously.
* Furthermore, learning representations which preserve cell property information is a application of obvious interest to the ICLR community.
* This paper performs the right experimental evaluations for its method, starting with a synthetic dataset, then moving to a real dataset where ground truth cell type information is available, and finally a real dataset with only brain region information, and compares against appropriate competitor methods.

**Weaknesses:**

* The framing in terms of the "Platonic Representation Hypothesis" [1] feels like hype chasing. What is being presented here is just a contrastive learning method which is meant to identify similarity and differences in properties of neurons. This is not actually related to the PRH in a meaningful sense, which is about how representations _of the world_ converge across models in completely different domains. I am willing to raise my score if the framing of the paper (primarily title, abstract, introduction) is substantially revised to take this into account.
* The details of the method are not clear to me. CEBRA [2], at least as proposed, takes in a window of activity across a population of neurons (among other covariates) and maps it to a single latent point. I believe that here CEBRA is being applied to the activity of single neurons and their covariates but this is an important distinction which is not made explicit in the text.
* I am not convinced that other methods are being fairly compared to NeuPIR. There is a lack of detail about how hyperparameter selection occurred in the experiments which makes this difficult to evaluate. For instance, LOLCAT fails to label any neurons at all as Sst in Table 2 which suggests it wasn't tuned correctly for the task.
* The method is not significantly original as it is combining the pre-existing CEBRA architecture [2] with the VIC contrastive loss [3], making this nearly a pure applications paper. This is not necessarily a flaw but does put the burden of innovation on the value of its scientific findings.

[1] Huh, Minyoung, et al. "The platonic representation hypothesis." arXiv preprint arXiv:2405.07987 (2024).
[2] Schneider, Steffen, Jin Hwa Lee, and Mackenzie Weygandt Mathis. "Learnable latent embeddings for joint behavioural and neural analysis." Nature 617.7960 (2023): 360-368.
[3] Bardes, Adrien, Jean Ponce, and Yann LeCun. "Vicreg: Variance-invariance-covariance regularization for self-supervised learning." arXiv preprint arXiv:2105.04906 (2021).

**Questions:**

* The abstract says "PRH posits that representations of different activity segments of the same neuron converge, while segments from inherently dissimilar neurons diverge" (lines 19-21). What representations this is referring to is not clear in context.
* The introduction should be more specific about what the method actually is, what the contrastive objective is, etc.
* Identifying cell type seems like a paradigmatic task where compressing neural activity into binned firing rate loses important information which may be contained in the spike train.
* The method section should provide more information about what CEBRA is.
* The cross-animal generalization experiment (5.3) is interesting but as predicting cell type is the more relevant problem I would be interested to see a similar generalization experiment with a cell type dataset as in 5.2.

---

> ### Author Response · Authors · 2024-11-14
>
> The abstract says "PRH posits that representations of different activity segments of the same neuron converge, while segments from inherently dissimilar neurons diverge" (lines 19-21). What representations this is referring to is not clear in context.
>  - PRH inspired us to treat each neuron as a system and learn the time-invariant intrinsic properties of each neuron's system representation. Our representation indeed differs from the Platonic representation, as you pointed out. Using "Neuron Platonic Intrinsic Representation" is not appropriate, and we have revised the title accordingly. Additionally, we have clarified the relationship between PRH and Neuron Intrinsic Representation in both the abstract and the introduction.
>
> The introduction should be more specific about what the method actually is, what the contrastive objective is, etc.
> - In this paper, we set the optimization objective for obtaining the intrinsic representation of neurons as follows: clips from the same neuron should have a higher average similarity than clips from different neurons. We employ a contrastive learning approach to achieve this optimization goal, treating different segments from the same neuron as positive pairs and segments from different neurons as negative pairs. During the implementation of the contrastive learning method, we realized that directly optimizing for the separation of samples from different neurons might be too rigid (since different neurons do not necessarily mean dissimilarity). Therefore, we chose the VICReg approach, which only uses positive pairs for optimization but indirectly separates dissimilar samples through regularization terms.
> Thank you for your suggestion. We have revised the introduction to make the reasoning clearer and to provide more detailed key points.
>
> Identifying cell type seems like a paradigmatic task where compressing neural activity into binned firing rate loses important information which may be contained in the spike train.
> - Identifying cell types is indeed a paradigmatic task, but our work is not focused on cell type identification. Rather, we use this as a downstream task to demonstrate that the learned time-invariant intrinsic representations contain meaningful information, of which cell type is one component. Although binning the firing rate may lose some detailed information, it still retains key features of neuronal activity. For example, binning the firing rate allows us to observe the overall response pattern of neurons under specific stimuli. What we are learning is the time-invariant intrinsic representation of individual neurons, which is also a overall representation. Binned firing rate is a standard preprocessing step in neuroscience data analysis when processing large-scale neural data. This approach is primarily aimed at reducing computational complexity and data storage requirements, making some level of information compression necessary. On the other hand, it aligns with the fact that neurons encode information according to frequency patterns.
>
> The method section should provide more information about what CEBRA is.
> - CEBRA serves as an encoder for the dynamic activity information of neuronal populations. Its advantage lies in its ability to encode both neuronal activity data (N*T) and corresponding auxiliary variables, such as behavior and external stimuli (M*T), into a low-dim (D*T). In CEBRA, D does not correspond to individual neurons, aiming to uncover the relationships between neuronal activity and these variables.
> Our work focuses on obtaining intrinsic representations for individual neurons. We cleverly utilize CEBRA from a different perspective as a preprocessing step for our input data. From each neuron, we use CEBRA to integrate the single neuronal peripheral information (such as activity of neighboring neuronal populations and behavioral data) for each segment of a single neuron. This process encodes the peripheral information associated with each segment of an individual neuron. Our work primarily involves contrastive learning to different segments of the same neuron.
> Thank you for your suggestion. I have added a detailed description of CEBRA in the Methods section, along with an explanation of how we use it.
>
> The cross-animal generalization experiment (5.3) is interesting but as predicting cell type is the more relevant problem I would be interested to see a similar generalization experiment with a cell type dataset as in 5.2.
> - Thanks for the suggestion, the cross animal results are more meaningful for neuroscience, so we changed the evaluations from one animal to multiple animals. Specifically, we have four mice, so 4-fold was used, with the folds based on the identity of the mice. We also divided a validation set and used a random search strategy to tune the hyperparameters of methods being compared, employing the best models on the test set. We have revised the result in the paper.

---

> > ### Author Response · Authors · 2024-11-14
> > **About the novelty**
> >
> > CEBRA is just our data preprocessing step, and CEBRA itself is used to study neuronal populations.
> > We consider the problem from the perspective of a single neuron and cleverly use CEBRA to process the surrounding information relative to a single neuron. CEBRA is equivalent to a step in data preprocessing.
> > It is unfair to regard our work as A+B work. We are the first to consider representation learning of the intrinsic properties of single neurons from a contrastive learning perspective, which is is a whole new field of computational neuroscience.
> > We sincerely hope you will consider our work again, thank you!

---

> > ### Comment · Reviewer_bVqe · 2024-11-14
> >
> > Thanks for the thoughtful response. In light of the updated experiments and the new abstract I'm happy to increase my overall score to a 5 and soundness to 3. Could you address the other weaknesses I mentioned? In particular, if major changes have been made to the text in the method or introduction section, it would helpful to delineate them. (Also, note that the method's name does not currently appear throughout the paper -- perhaps a LaTeX issue?)
> >
> > I stand by my assessment of the novelty of the proposed work -- this is an applications paper. That said, the specific approach of contrastive learning for the broader research direction of forming data-based representations of individual neurons is new to my knowledge as well as a natural pairing of problem and technique, and the scientific findings have merit.

---

> > > ### Author Response · Authors · 2024-11-14
> > >
> > > Thanks for the reminder, I've fixed the method‘name and reuploaded the pdf
> > >
> > > In addition, I also reorganized the introduction according to the ideas you suggested, and told it in the following order:
> > > (1) Try to obtain the intrinsic representation of neurons inspired by Platonic representation.
> > >
> > > (2) Contrastive learning is the idea of ​​​​obtaining the inherent representation of neurons.
> > >
> > > (3)VICReg is the specific implementation method.
> > >
> > > (4) The data preprocessing problems faced when contrastive learning is directly applied are solved by the CEBRA method.
> > >
> > > (5) Brief summary and discussion of various experiments.
> > >
> > > The methods section reexplains how we use CEBRA to avoid misunderstandings.
> > >
> > > Your suggestions have greatly improved the readability and clarity of our papers. Thank you sincerely!

---

> > > ### Author Response · Authors · 2024-11-17
> > > **Friendly Reminder: Review Feedback Update**
> > >
> > > Dear Reviewer bVqe,
> > >
> > > I hope this message finds you well. I would like to sincerely thank you for your constructive feedback and for your support in improving my submission. It truly means a lot to me.
> > >
> > > According to your suggestion, I have rewritten the Introduction and further explained the CEBRA content in the method part. Figure 1 has also been redrawn to help better understand the method. I've uploaded the revised PDF and hope you can check it again.
> > >
> > > I noticed that the review system hasn't reflected the score adjustment. If it's not too much trouble, I would greatly appreciate it if you could update the score at your convenience.
> > >
> > > Thank you again for your time and consideration!
> > >
> > > Best regards!

---

> > > > ### Comment · Reviewer_bVqe · 2024-11-18
> > > >
> > > > And I would like to sincerely thank you for a cordial and productive rebuttal phase! The description of the method is much clearer now. I'm raising the presentation to 3 and overall score to 6.

---

### Official Review · Reviewer_mMop · 2024-11-04

**Soundness:** 3
**Presentation:** 2
**Contribution:** 3
**Rating:** 6
**Confidence:** 5

**Summary:**

This work proposed NeurPIR, which focus on learning a platonic representation from neural activities data to reflect the inherent properties and neuronal identity, relating to molecular information. The goal of this work is to learned representations robust to variations due to external stimuli and experiment conditions. It utilized the self-supervised multi-segment contrastive learning strategy from CEBRA, and learning representations for neurons with compare data from different segments, different behavior information, session information, and for neurons share similar functional roles to align closely in their representations. And they further aggregate the representation with adaptive average pooling to extract time-invariant representations. They further incorporate VICReg loss to enhance the prediction. The work is evaluated on three benchmarks: Izhikevich simulation model, spatial transcriptomics data with neural activities, neuron location with out-of-domain data, and compared with two existing baselines NeuPRINT and LOLCAT.

**Strengths:**

1. The work is evaluated on three representative benchmarks, including one synthetic dataset, and two neural datasets. And it is compared against with two baselines and demonstrating SOTA performance in most of the tasks.
2. The work utilized a novel, distinct and effective method to utilize contrastive learning strategy to learn the platonic representation, compared to NeuPRINT with learning time-invariant representations with a neuron-wise look-up table for dynamics forecasting during self-supervised learning, or LOLCAT with label-guided representation with end-to-end supervised learning.
3. The out-of-domain evaluation on unseen mice is an important question, which increases the soundness of the evaluations.

**Weaknesses:**

1. The core of the proposed approaches similar strategy as CEBRA to utilize the contrastive self-supervised approach for representation learning. The major difference from CEBRA is that it utilizes the adaptive average pooling to aggregated into time-invariant embedding, which might not necessarily guarantee the converge of the representation based on the data sampling. Further investigation could be done on how to guarantee converge based on different sampling strategy, or the requirement of amount of data to affect the predictive performance of the downstream tasks.
2. The evaluation on Steinmetz dataset is evaluated on decoding brain region with the learned intrinsic representations is related to analyze the invariant properties of the neurons, while brain region is only coarsely reflecting information from neuronal level, would it be possible to evaluate on more fine-grained information such as spatial location of individual units?
3. Ablation studies of VICReg loss should be performed.
3. Sensitivity analysis (i.e. error bar) are not included in Fig 4, the effect of data shuffling, random initialization, etc. could be reported.
4. Figure quality (i.e. font size, resolution) in the paper as limited, presentation and writing could be improved.

**Questions:**

1. Time complexity of NeuPIR compared to other baselines?
2. How many data samples are required to learn effective embedding?

---

> ### Author Response · Authors · 2024-11-14
>
> Time complexity of NeuPIR compared to other baselines?
>
> - If we disregard the time spent on data preprocessing, the time consumed by all methods would indeed increase proportionally with the multiple of the sample size. You might have doubts about this statement, especially when we employ contrastive learning, as it should not exhibit proportional growth. However, it is important to note that we are using VICReg, which only requires the training of positive sample pairs. About time cost, Neuprint takes the representation of neurons as an input embedding of a transformer and updates it using a backpropagation algorithm, which is difficult to train, takes 25 hours, takes up 60 gigabytes of memory on top of the A100(80G). LOLCAT is a supervised training, it is the easiest to converge, the same A100(80G) above, takes 1 hour, uses 10G memory. Our method is a self-supervised contrast learning process, which is the same as A100(80G), takes 15G memory and takes 4h to converge.
>
> Would it be possible to evaluate on more fine-grained information such as spatial location of individual units?
>
> - We can conduct more fine-grained brain area classification experiments, but if the brain areas are divided more finely, the number of samples in each brain area may not be sufficient, and the neurons in some brain areas are very similar, making it difficult to distinguish. These are all worth exploring in the future, thank you for the new inspiration
>
> How many data samples are required to learn effective embedding?
>
> - each session we use is 5000*400 (timepoints*neurons), each neuron we will sample 8 positive pairs, effective embedding needs at least two session like this.
>
> We have revised the paper to do K-fold cross-validation, and each fold is experimented with with a different random number and reports the mean and variance. We tried to delete any part of VICReg for experiments, but no convergence could be achieved if any part was deleted.

---

> > ### Comment · Reviewer_mMop · 2024-11-25
> >
> > Thank you for your response and for addressing some of the concerns. I have decided to maintain my current score.

---

### Official Review · Reviewer_eGKZ · 2024-11-04

**Soundness:** 2
**Presentation:** 2
**Contribution:** 3
**Rating:** 6
**Confidence:** 4

**Summary:**

The paper introduced NeurPIR, a self-supervised contrastive learning approach to learn intrinsic representation for each neuron from population dynamics. The method leveraged CEBRA and VICReg for representation learning, and was evaluated using synthetic and two mouse datasets, showing ability to learn representations indicative of neuronal intrinsic properties that are decodable by downstream classifiers.

**Strengths:**

* The method combines CEBRA and VICReg to incorporate surrounding information and learn enhanced representation with contrastive learning, which is a novel approach.
* The paper is well motivated and tackles an important problem in neuroscience.
* The writing is clear and logical.

**Weaknesses:**

* The usefulness of the out-of-domain evaluation on Steinmetz dataset is questionable. It seems as described on line 266, the self-supervised contrastive learning is performed on all neurons of all mice, including the test mice, i.e. the self-supervised model and classifier have to be retrained everytime new mice come in. Can any part of the model at least be reused during test time?
* Model architecture is not clearly explained by figures or texts. Some details of the methods are missing (elaborated in Questions).
* Some ablation studies are missing that would otherwise be helpful to understand the method in greater detail. For example, an ablation on which surrounding information included in the CEBRA framework has the most impact, or an ablation on different choices of contrastive learning methods besides VICReg might be helpful.
* Results in tables and figures do not have errorbars. Adding sensitivity analyses would be helpful to quantify how significant the improvements of NeurPIR over the baselines are.

**Questions:**

* Line 54 and 55: the paper is motivated to make segments of activity from the same neuron or similar neurons to converge, while dissimilar neurons diverge. However, VICReg only uses positive pairs, and adds regularization to prevent representation collapse. How does using VICReg help push dissimilar neurons apart according to the motivation?
* Figure 1: description in the texts and accompanying caption to understand this figure are missing. The figure does illustrate negative pairs, however, VICReg does not use negative pairs as mentioned above. How are negative pairs processed by the NeurPIR model?
* Line 134: the goal was to learn intrinsic neuronal representations on neuron population data, but it does not seem that activity of other neurons are used in the model (equations 1 to 4). How does the model use population dynamics to learn neuronal representations?
* Line 149: what is the length of one segment? Is there a chance that two randomly selected segments overlap with each other?
* Line 155: what is session information $X_{se}$? What are dimensions of $X_{st}$, $X_{be}$, $X_{se}$, $X_{si}$?
* Line 161: can the author provide more details on what is adaptive average pooling?
* Line 202: how the target value $\mu$ was set?
* Figure 4: how about precision, recall, and F1 scores (to be consistent with Tables 1 and 2)?
* It would also be helpful to provide additional details on the architecture design and training process, e.g. hyper-parameters, training time, etc.

---

> ### Author Response · Authors · 2024-11-14
>
> Line 54 and 55:
> - Although VICReg does not explicitly use negative pairs like other contrastive methods, it still indirectly promotes the separation of different samples through the variance and covariance regularization terms. The variance regularization ensures that the representations have sufficient spread, meaning that for dissimilar samples, their feature representations are less likely to collapse into a small region of the embedding space. The covariance regularization helps ensure that features are not highly correlated, promoting more distinct and diverse representations. VICReg relies on a more subtle approach, achieving the desired separation through variance and covariance regularization.[1]
>
> Figure 1:
> - In this paper, we set the optimization objective for obtaining time-invariant intrinsic representations of neurons as follows: clips(segments) from the same neuron should have a higher average similarity than clips from different neurons. Figure 1 illustrates how this objective can be achieved using a contrastive learning approach, where different clips from the same neuron are treated as positive pairs and clips from different neurons are treated as negative pairs. This is the main idea conveyed by Figure 1.
> However, when selecting a specific contrastive learning method, we realized that directly optimizing for the separation of clips from different neurons may be too rigid (as different neurons do not necessarily represent dissimilarity). Therefore, we chose to implement VICReg, which optimizes using only positive pairs, but indirectly separates dissimilar samples through regularization terms. We have revised the caption of the figure to explain more detail.
>
> Line 134:
> - In this paper, we are not learning representations of neuron populations, nor are we performing dimensionality reduction on neuronal population activity. Instead, we aim to learn a time-invariant intrinsic representation for each individual neuron. Each neuron is considered separately, and the activity information from the remaining neurons in the population is treated as peripheral information (equation 1: X_st) for that specific neuron, which acts as auxiliary variables. The peripheral information for each neuron is processed and encoded using CEBRA.
>
> Line 149:
> - If a neuron has data from multiple sessions, we will select segments of 512 time points from different sessions as positive pairs for that neuron. If a neuron only has data from a single session, we will randomly select non-overlapping segments, with lengths randomly ranging from 200 to 512 time points, as positive pairs for that neuron. During implementation, we have ensured the use of an algorithm that prevents overlap between these segments.
>
> Line 155:
> - A session refers to a single stage or individual experiment in a neuroscience study where data is collected. If you want to know more about session, please read this link[2]. The input dimension of the model is the same as the length of the segment. For example, if there are 300 time points, the first session's Xse would be represented as [1, for i in range(300)].
>
> Line 161:
> - Based on the previous question, if a neuron only has data from a single session, we will randomly select non-overlapping segments of lengths ranging from 200 to 512 time points as positive pairs for that neuron. For these segments of varying lengths, we ultimately need to map them to the same dimensionality. Adaptive average pooling is a suitable tool for this task, as it does not require explicitly specifying a fixed pooling window size. Instead, we can set a consistent output size, and the pooling operation will automatically adjust to accommodate the varying segment lengths. Adaptive average pooling is capable of handling inputs of different lengths and extracting important feature information, mapping them to a consistent dimensionality.
>
> Line 202:
> - It is set to a fixed value of 1, as we referenced in the original VICReg paper and other works that utilize VICReg, which all adopt this setting. The choice of the target value is an interesting topic worth further exploration, especially in terms of its impact on negative pairs. However, this would become the focus of another paper dedicated to contrastive learning.
>
> Figure 4:
> - We replaced the original bar charts with a table to present more detailed information, including the precision, recall, and F1 scores details.
>
> [1] Bardes, Adrien, Jean Ponce, and Yann LeCun. "Vicreg: Variance-invariance-covariance regularization for self-supervised learning." arXiv preprint arXiv:2105.04906 (2021).
> [2]https://allensdk.readthedocs.io/en/latest/visual_behavior_optical_physiology.html#session-structure

---

> ### Author Response · Authors · 2024-11-15
> **response weeknesses**
>
> - Our model is a two-stage design: the first step: the process of obtaining neuronal representations is a self-supervised process, sharing a single model, and any new mouse data can be added to the old data for training. Step 2: The training set and test set are presented in the second step, and K-Fold is used for downstream tasks on the neuronal representations obtained in the first step, and folds along the mice identity.
> - We enriched the content of Figure1 and wrote a detailed caption. The revised pdf has been uploaded.
> - The role of CEBRA is to integrate the surrounding information of a single neuron, and if the surrounding information is not integrated, the model cannot converge if it is directly input, which is what we did at the beginning, and then we found CEBRA to solve it.
> Other contrastive learning methods require the design of negative sample pairs, which is relatively complex for neuroscience data, and the design of negative sample pairs has a greater impact, while VICReg does not need to design negative sample pairs
> - We performed a k-fold experiment with three different random numbers for each fold, and the final results were reported as mean and standard deviations, and the results table has been updated.
>
> Thanks for your advice. I hope you can consider it again!

---

> ### Comment · Reviewer_eGKZ · 2024-11-28
>
> Thank you for your response. That resolved most of my questions. I'm still a bit unclear on how NeurPIR incorporates activity from other neurons. $X_{st}$ in Equation 1 represents the visual stimuli, so it seems actual activity of other neurons in the population is not included as inputs to NeurPIR. Could you clarify on this point and also provide the dimensionality of each component $X_{st}$, $X_{be}$, $X_{se}$, $X_{si}$?
>
> Minor points: the revised manuscript looks better than the original; however, some typos and formatting issues still remain in this version:
>
> * Some of the terms are used inconsistently, such as NeuPRINT (sometimes written as NeuPrint, Neuprint, NeurPrint), Lamp5 (sometimes written as LAMP5), Vis, Thal, Hipp, Mid (written as vis, thal, hip, mid on line 425), $X_{st}$, $X_{be}$, $X_{se}$, $X_{si}$ in Figure 1 should be properly subscripted.
> * Rows/columns in tables should be in increasing order of average performance.
> * Typos on line 424 (should be "LOLCAT and NeuPRINT"?) and 428 (sentence missing subject)

---

> > ### Author Response · Authors · 2024-11-28
> >
> > Thank you for your valuable feedback on our work.
> > - $X_{st}$ consists of two components: peripheral neuronal activity and visual stimulation. The dimensions of $X_{st}$ are $n+1$: The $n$ dimensions are the activity of the surrounding neurons relative to the neuron you're considering, because there will be coordinates for each neuron in the dataset, so for each neuron, we'll take the 47 nearest neurons in the experiment. The last dimension is visual stimuli, and we have a relatively simple way of dealing with visual stimuli: Blank visual stimuli are represented by 0, Drifting Grating visual stimuli by 1, Natural Scenes visual stimuli by 2, and each time point is represented by a fixed number. In the future, we could try to encode the video of visual stimuli so that it changes over time, but we currently only encode it according to the type of visual stimulus.
> > - The dimension of $X_{be}$ is 1: running speed each time point.
> > - The dimension of $X_{se}$ is 1: session number
> > - The dimensions of $X_{si}$ are 10: The total dimension of $X_{be}$, $X_{se}$ and $X_{st}$ is 50, we use CEBBRA project them to the low dimension 10.
> >
> > About Minor points: We have fixed these bugs, but November 26th is the Last day that authors may upload a revised PDF. I promise that once the paper is accepted, I will upload the revised pdf on arxiv for other readers to read clearly.
> >
> > We have made significant revisions based on your comments, and we believe these changes have improved the quality of the work. We respect your assessment, and we still hope you will consider the updates favorably. We would greatly appreciate a reconsideration of the final score.

---

> > > ### Comment · Reviewer_eGKZ · 2024-11-28
> > >
> > > Thank you for the clarification. I believe you can still upload the PDF today (Nov 27). In any case, I have raised my rating accordingly.

---

### Author Response · Authors · 2024-11-25
**Summary and Sincere Gratitude for Reviewers‘ Constructive Feedback During the Rebuttal**

Dear Reviewers,

I would like to express my heartfelt gratitude to each of you for the time, effort, and valuable feedback provided during the rebuttal period. Your constructive comments have been instrumental in refining the manuscript and enhancing its overall quality.

We have carefully considered all the points raised, we have made several significant revisions to the manuscript, including:

- Rewriting the Abstract and Introduction: We have clarified and strengthened both sections to better highlight the main contributions and context of the work.
- Enhancing the quality and content of the figures, improving both their visual clarity and the depth of information conveyed.
- Adding Necessary Experiments: We have incorporated additional experiments as suggested, providing further validation and support for our findings.
- Adjusting the Formatting: We have carefully revised the manuscript to ensure it fully complies with the formatting requirements of ICLR.

Your dedication to improving the quality of the work is evident, and it has been a privilege to engage with such a knowledgeable and thorough review process. I believe that your feedback has not only improved this submission but also contributed to making the ICLR review process better and more meaningful.

---

### Meta-Review · Area_Chair_sy9e · 2024-12-10

**Metareview:**

This paper introduces a novel technique for self-supervised learning on neural data based on the idea that there is an "intrrinsic representation" for a neuron that reflects its underlying physiological properties, and which should be identifiable from distinct bouts of activity at different times or experimental conditions. Based on this, the system employs contrastive learning to match representations for different bouts of activity from the same neuron, and push apart representations for different neurons. The authors claim that once trained the representations that are learned do indeed capture critical features of the neuron, such as its cell type or underlying physiological parameters. They support these claims with a set of experiments on simulated data and another on real data with cell-type annotations.

The strengths of this paper are its clarity, well-articulated motivations, and convincing empirical demonstrations. The weaknesses are that it overstretches the anaology to the Platonic Representation Hypothesis, and was missing some important ablation studies. These weaknesses are partially addressed in the updated manuscript (as are the other comments of the reviewers), so coupled with the clear strengths of this paper, a decision of accept (poster) was reached.

**Additional Comments On Reviewer Discussion:**

The reviewer discussion was courteous and productive. The authors attempted to address the reviewer concerns, and the reviewers engaged them in discussion. In the end, the scores all came up above the acceptance threshold. All reviewers were considered equally in coming to this decision.

---

### Decision · Program_Chairs · 2025-01-22

Accept (Poster)